# mCrutch: A Novel m-Health Approach Supporting Continuity of Care

**DOI:** 10.3390/s23084151

**Published:** 2023-04-21

**Authors:** Valerio Antonio Arcobelli, Matteo Zauli, Giulia Galteri, Luca Cristofolini, Lorenzo Chiari, Angelo Cappello, Luca De Marchi, Sabato Mellone

**Affiliations:** 1Department of Electrical, Electronic and Information Engineering (DEI), Alma Mater Studiorum, University of Bologna, Viale Risorgimento 2, 40136 Bologna, Italy; 2Department of Industrial Engineering (DIN), Alma Mater Studiorum, University of Bologna, Via Umberto Terracini 24-28, 40131 Bologna, Italy; 3Health Sciences and Technologies-Interdepartmental Center for Industrial Research (CIRI-SDV), Alma Mater Studiorum, University of Bologna, 40136 Bologna, Italy

**Keywords:** crutches, gait monitoring, telerehabilitation, mobile-health, instrumented walking aids, wireless sensors

## Abstract

This paper reports the architecture of a low-cost smart crutches system for mobile health applications. The prototype is based on a set of sensorized crutches connected to a custom Android application. Crutches were instrumented with a 6-axis inertial measurement unit, a uniaxial load cell, WiFi connectivity, and a microcontroller for data collection and processing. Crutch orientation and applied force were calibrated with a motion capture system and a force platform. Data are processed and visualized in real-time on the Android smartphone and are stored on the local memory for further offline analysis. The prototype’s architecture is reported along with the post-calibration accuracy for estimating crutch orientation (5° RMSE in dynamic conditions) and applied force (10 N RMSE). The system is a mobile-health platform enabling the design and development of real-time biofeedback applications and continuity of care scenarios, such as telemonitoring and telerehabilitation.

## 1. Introduction

### 1.1. Background

Gait, or the act of walking, is a fundamental human function. When an individual’s ability to walk is compromised due to an impairment or injury, rehabilitation efforts often prioritize the restoration of this ability [1]. Approximately 10% of adults experience limitations in their mobility or balance as a result of conditions affecting the central nervous system (CNS). Gait impairments are the result of a variety of medical conditions, including lesions of the central and peripheral nervous systems. Walking aids are frequently prescribed to enhance mobility and balance [2], and are also prescribed for neurological patients [3]. Other common medical conditions are osteoarthritis, multiple sclerosis, fractures, muscle lesions, and stroke [4]. Gait impairments have a significant impact on patients’ quality of life [5], and walking aids play a major role in increasing their daily mobility and independence [6]. Crutches are the most common assistive devices for individuals with mobility impairments [7]. As the number of individuals with mobility impairments increases with the longer life expectancy, it is likely that the use of assistive devices, such as crutches, will also increase over time [8,9].

Whenever possible, the use of crutches in place of wheelchairs is often preferred by clinicians because they promote a patient’s participation, allow them to keep an upright posture with related physiological benefits, and improve their independence in daily living activities. These factors can lead to better long-term recovery outcomes. In addition, crutches may be a more practical and convenient option for individuals who only require short-term or intermittent mobility assistance [10,11].

From a biomechanical point of view, crutches increase an individual’s base of support while walking. They also allow patients to transfer part of their body weight from their lower to upper limbs while walking or standing [12].

There are three main types of crutches, as illustrated in Figure 1: axillary (or underarm), forearm (or elbow/Canadian/Lofstrand), and gutter crutches. Each type has its own unique characteristics and may be suitable for certain individuals depending on their specific needs.

Several different walking patterns can be adopted when using crutches. One common pattern is the two-point crutch gait, where the crutches and the affected limb move forward together while the weight-bearing limb follows when the crutches are on the ground. Another pattern is the two-point gait, where the crutches and the non-affected limb move forward in alternating steps. The four-point gait pattern is characterized by a coordinated movement of the crutches and the non-affected limb. Finally, in the step-to-gait pattern, the affected limb moves forward, followed by the non-affected limb, while the step-through gait pattern shows the opposite sequence [7,13,14,15,16].

During rehabilitation sessions, therapists often do not have the tools for a quantitative assessment of patients’ progress and usually rely on visual observation and clinical scales that cannot capture relatively small changes [17,18]. This can lead to an incorrect share of the weight between the crutches or incorrect crutch positioning while walking or standing, especially in unsupervised settings. Sensorized crutches, embedding sensors, processing capabilities, and connectivity options can address the need for real-time and long-term monitoring of crutch usage, providing therapists with more accurate and quantitative information on patients’ recovery [19].

### 1.2. Related Work

Sensorized crutches, or instrumented crutches, are assistive devices developed to support therapists and patients in rehabilitation programs. Crutches are equipped with sensors that measure various quantities, such as load, acceleration, and orientation. The data collected by the sensors are used to provide feedback to both the patient and the therapist, enabling a more accurate assessment of the patient’s progress and supporting clinical decisions. State-of-the-art, in the field of sensorized crutches, includes a range of different designs and features.

Merrett et al. [20] designed an instrumented crutch for monitoring a patient’s weight bearing during the recovery process in the field of orthopedic rehabilitation. The system aimed to: (i) monitor the force applied in the direction of the crutch axis by means of a FlexiForce^®^ force sensitive resistor mounted inside the crutch; (ii) measure the tilt of the crutch using a tri-axial accelerometer; and (iii) determine the grip force applied to the handle using a membrane potentiometer.

Sardini et al. [21] developed wireless instrumented crutches that utilize three strain-gauge bridges to measure axial and shear forces and a tri-axial accelerometer to measure anterior–posterior and medio–lateral angles. The crutch embeds a conditioning and transmission circuit and a battery power management circuit. Data are transmitted wirelessly via Bluetooth to a computer. Crutches were tested on one subject and later used in a study conducted by Tamburella et al. [22], where it was found that subjects who received load auditory feedback while using sensorized crutches had significantly higher usage time compared to subjects who did not receive any feedback. The result of this study suggests that auditory feedback may be an effective method for improving crutch usage in subjects with central nervous system lesions.

The GCH System 2.0 is an updated version of the prototype developed by Chamorro-Moriana et al. [23]. The device embeds a microcontroller that collects data from a cylindrical load cell placed at the tip. Unlike the previous version, all of the electronics of the GCH System 2.0 are contained inside the shaft of the crutch. The device also includes dedicated software that runs on a computer and provides the user with weight-bearing information and visual feedback [24]. The prototype was tested with a cohort of 10 patients to evaluate the use of various feedback modalities, including visual, acoustic, concurrent, terminal, and descriptive feedback, to improve load–balance distribution between crutches and gait patterns in indoor applications, such as in clinics and laboratories [25].

Narváez et al. [26] presented a prototype of instrumented crutches to monitor patients’ gait patterns. The prototype utilizes strain gauges inside the handle, measuring the applied force, and an IMU for estimating the orientation of the crutch. Data was collected and transmitted via a Bluetooth module to a laptop. The authors segment the walking data into the swing and stance phases and use the characteristics and distribution of such phases to identify different gait patterns automatically.

Brull et al. [27] developed a sensorized tip for assistive devices, such as crutches or walking sticks, for monitoring individuals with multiple sclerosis. The tip was equipped with pressure sensors and a wireless module and could transmit the collected data to a computer in real-time. Data was used to calculate various parameters, such as the weight-bearing ratio and gait cycle. The sensorized tip was tested by patients with multiple sclerosis. The continuous monitoring of gait characteristics allows an accurate assessment of the patient’s state and disease progression. They also used their instrumented tip for monitoring the patients’ activities of daily living (ADL) using a classifier based on neural networks. The classifier was validated on a cohort of 13 volunteers performing four typical ADLs (walking, standing still, and going up and down the stairs) with an overall accuracy of 95% [27,28]. 

Sesar et al. [29] also developed an instrumented crutch tip for monitoring the force and pitch angle during gait rehabilitation. The tip, which can be attached to any crutch, embeds a two-axis inclinometer, a tri-axial gyroscope, and a force sensor to measure the force applied to the crutch and the pitch angle. A novel algorithm for estimating the pitch angle was also presented. 

Another possible application of instrumented crutches is in combination with lower-limb exoskeletons [30,31]. 

### 1.3. Motivation

The growth of wearable technologies has led to a proliferation of smart devices in various fields, including healthcare. The World Health Organization (WHO) defines “mHealth” as “the use of mobile devices, such as mobile phones, patient monitoring devices, Personal Digital Assistants (PDAs), and other wireless devices, for medical and public health practice” [32]. To our knowledge, there are no applications of sensorized crutches for the long-term monitoring of patients with motor impairments. We believe that a mobile health application based on a set of instrumented crutches could be a valid support to continuity of care scenarios for multiple reasons: Instrumented crutches can provide continuous, real-time monitoring of the patient’s mobility and gait pattern, enabling the objectification of the rate of progression of the rehabilitation for each individual patient.Instrumented crutches can collect and transmit data to smartphones or other mobile devices. The computational capacity of modern devices enables real-time applications/feedback and advanced reporting functions for both therapists and patients.Instrumented crutches in an mHealth scenario can improve the communication between the patient and the therapist, enabling remote monitoring and teleconsultation. Telerehabilitation applications can be important for patients living in underserved or remote areas. They would improve access to rehabilitation services and reduce the healthcare system’s burden.Through smart biofeedback applications and personalized reporting functions, instrumented crutches can empower patients and allow them to take a more active role in their rehabilitation program.

The COVID-19 pandemic highlighted the need for more efficient, accessible, and pervasive healthcare approaches, such as intelligent systems and mHealth technologies [33,34,35,36,37,38,39]. The use of sensorized crutches in an mHealth scenario supporting the continuity of care may also help reduce healthcare costs [40]. However, despite the variety of applications and prototypes reported in the literature, there is still the need to assess the potential impact of instrumented crutches in the continuity of care applications, from real-time guidance in rehabilitation facilities to long-term remote monitoring outdoors and at home.

The aims of our study are as follows:To develop a set of instrumented crutches suitable for mobile health applications. Expected outcomes are orientation angles and applied loads.To develop a smartphone app, mCrutch, for the management of the instrumented crutches and for enabling real-time applications.To verify the accuracy of the estimate for the orientation angles and the applied loads.To keep manufacturing costs in line with those of mass-market technologies.

## 2. Materials and Methods

A pair of Lofstrand (or Canadian) crutches was instrumented to measure the axial force and orientation. A novel dedicated Android application, mCrutch, processes, visualizes, and stores data collected by the instrumented crutches in real-time (Figure 2). All of the electronics were contained inside the crutch’s original structure.

Each crutch embeds an Arduino RP2040 equipped with a 6-axis inertial measurements unit (IMU). The Arduino was also connected to a uniaxial load cell placed at the tip of the crutch inside a custom mechanical structure. 

Figure 3 shows the crutch prototype and lists the different components. 

### 2.1. Electronic Components

Most of the electronics were inserted in the crutch’s handle cavity, approximately 84 mm deep, with a 28 mm diameter. The handle cavity communicates with the crutch’s vertical shaft and the forearm cuff (Figure 3), enabling the run of wires inside the crutch structure. Figure 4 shows all the electronic components: the main boards (U2) and the power management board (U3). U2, in Figure 4, is an Arduino Nano RP2040 Connect [41], selected for its small form factor (18 × 45 mm) that can fit the housing in the crutch’s handle; the Arduino board collects data from the embedded IMU, and the load cell and manages the WiFi connection with the Android app. The main board hosts a Raspberry Pi RP2040 microcontroller unit (MCU), a U-blox Nina W102, an STMicroelectronics LSM6DSOX 6-axis IMU with an on-chip temperature sensor, and an MP34DT05 microphone. U3, in Figure 4, is the power management board. It features an MCP73832 single-cell charger, and it was connected to a lithium-ion (Li-Ion) battery (U1).

With reference to Figure 4, the full list of components is provided below:U1—Power supply: Li-Ion battery RS-ICR14500 [42], 3.7 V at 820 mAh.U2—Processing, data acquisition and wireless communication management: Raspberry Pi RP2040 MCU [43], a dual-core 32-bit ARM Cortex operating at a frequency of up to 133 MHz, 264 KB on-chip SRAM, up to 16 MB off-chip Flash and various digital and analog peripherals (SPI, I2C, UART, ADC, etc.).U2—Wireless communication: U-blox Nina W102 [44], Bluetooth V4.2, and WiFi 802.11 b/g/n module.U2—IMU: LSM6DSOX [45], STMicroelectronics micro electro mechanical system (MEMS) sensor, which embeds a three-axial accelerometer and three-axial gyroscope (6-axis IMU) with a full-scale acceleration range up to ±16 g and a maximum angular rate of ±2000 dps. It is used to measure linear acceleration and angular velocity of the crutch for estimating its orientation.U3—Li-Ion on-board battery charger: MCP73832 [46], Microchip 500 mA linear charger management controller for single cell Li-Ion/Li-Polymer battery.U4—Voltage converter: ANGEEK DC-DC Step-Up, 0.9–5 V to 5 V, operating frequency 150 KHz, conversion efficiency 85%. It boosts Arduino 3.3 V output to 5 V to power the load cell (U5).U5—Load cell: uniaxial load cell FX293X-100A-0100-L [47], analog output (0.5–4.5 V) by TE connectivity, with a full-scale range of 500 N, a precision of ±0.25% FS and a round shape (diameter 19.7 mm, height 5.45 mm) used to measure axial force applied on the crutch tip.SW1—Slide switch to power ON/OFF the device.LD1—RGB LED, signals the system status (green: power on, blue: connected to the host device/smartphone).R1—Limits the current to LD1.R2, R3—Level shifter to adapt 0.5–4.5 V load cell output to RP2040 MCU ADC channel 0–3.3 V.

Table 1 the summarizes the features of the system.

Data collection from LSM6DSOX IMU was performed at 204 Hz with a ±4 g range for the accelerometer and at 208 Hz with a ±2000 dps range for the gyroscope. The orientation of the device was estimated with a sensor fusion approach based on Madgwick filter [48]. The choice of a high-performance MCU, such as the one on Arduino RP2040 Connect, provided a smooth data fusion with a high data rate of up to 100 Hz. The same rate was used to sample the output of the load cell, using one RP2040 ADC channel. Force signals were digitally filtered with an 8-point window moving average filter. Orientation and force data were transmitted to the smartphone via a WiFi protocol through the U-blox Nina W102 module. Therefore, the system sends in real-time, every 10 milliseconds (100 Hz), a frame consisting of the values of the triaxial accelerometer, the triaxial gyroscope, the estimated orientation angles, the applied load, the battery charge level, and the timestamp. The manufacturing cost for the current prototype version was approximately EUR 100 per crutch.

### 2.2. Smart Tip and Mechanical Structures

A custom mechanical structure, named smart tip, was designed to house the uniaxial load cell (Figure 4, U5 device) into the tip of the crutch. The smart tip ensures that: Only the component of force applied along the crutch shaft axis is measured. Other components of force (perpendicular to the shaft) and moments are removed by dedicated low-friction Teflon components mechanically insulating the miniaturized cell.When an external force is applied to the crutch, the measured force value reflects the applied load.

The smart tip structure was made of aluminum with Teflon elements (see Figure 5).

The plastic cases for the battery and the electronics in Figure 3 were made of thermoplastic polyurethane (TPU) and manufactured with a 3D printer.

### 2.3. The mCrutch App

The mCrutch system was designed to be connected to a host device running an Android operating system (a Samsung Galaxy A50 in this study). The mCrutch app was developed in Android Studio 2021.3.1 (Dolphin); its profiler was used for evaluating the computational cost of the app. This application is responsible for collecting, storing, and visualizing data. The current user interface was designed for researchers and software developers and only acts as a data collector. 

Android OS allows the creation of a WiFi hotspot, and the connection between the instrumented crutches and the host device was established through a TCP socket communication, where the mCrutch app acts as a server and the instrumented crutches act as clients. A unique SSID and password, encrypted using WPA2 encryption, were used for the data transfer. 

The data stream includes orientation and applied force but also information, such as the timestamp, the battery level (expressed as a percentage from 0% to 100%), battery status (either charging or not charging), and the status of the crutches (e.g., error messages). The estimate of the orientation through the Madgwick filter runs on the Arduino board [48]. 

The sampling frequency was set to 100 Hz, which is fast enough for most motion analysis applications. To verify data loss, a function on the smartphone app compares the current timestamp with the previous one. If there is any gap in the data, a message is sent to the user through the display of a string on the user interface panel, and a 1 s vibration is activated too. This allows the user to be aware of the data loss. The mCrutch system architecture, along with flowcharts for each single layer, is reported in Figure 6. 

The interface of the mCrutch app includes a control panel (Figure 7A) with a “Connect” button in the center, which allows the user to initiate the connection with the crutches. Once the connection is established, the indicators for the left and right crutch will turn green, indicating that the connection has been successfully established. The user can start the data stream by pressing the “Start” button, while a chronometer (Figure 7C) shows the elapsed time (Figure 7B). Data are transmitted from the crutches and displayed in the “Data Indicators” box (Figure 7D). The user has the option to activate a real-time chart by switching on the “PlotData” flag (Figure 7E). The box displays the pitch angle in degrees [°] and the applied force in Newtons [N]. The user can stop the data stream by pressing the “End” button. Collected data are stored in the internal memory of the smartphone and can be accessed by the user for offline analysis.

### 2.4. Calibration Procedure

Calibration is an essential step in analyzing kinematics and dynamics quantities. It is crucial to calibrate custom devices that include transducers embedded within a mechanical structure, as the structure itself may alter the physical characteristics or positioning of the transducers in a way that deviates from the manufacturer’s recommendations. For example, a 6-axis IMU embedded in an Arduino system refers to an internal local reference frame that may not correspond to the reference frame of the device. The calibration converts the measurements from the internal local reference frame of the sensor to the desired reference frame on the device. 

A movement analysis laboratory was used to calibrate the instrumented crutches. The laboratory was equipped with four Kistler force platforms and a BTS motion capture SMART-DX EVOs system [49], systems that can be considered the gold standard for movement analysis. The crutches have a sampling frequency of 100 Hz, while the force platforms and the motion capture system have a sampling frequency of 1000 Hz and 250 Hz, respectively. 

Figure 8 illustrates the setup that was used during the calibration procedure. Cameras were used to capture the movement of the instrumented crutches through four optical reflective markers fixed on the crutch. During the calibration, a random sequence of movements and applied loads were produced in order to capture a wide range of orientation and force values. 

Data from the motion capture system and the instrumented crutches were imported in MATLAB (R2022b) for offline analysis. Camera data were down-sampled and synchronized with the IMU signals. As shown in Figure 8, a cluster of markers was used, where: marker *mk0* was placed on the tip, marker *mk1* was placed on the shaft, marker *mk2* was placed at the intersection between the handle and shaft, and marker *mk3* was placed at the end of the handle. A cluster of four markers has been used for two reasons: (i) in future validations with end-users, guidelines for full body kinematics would require a cluster of four markers on each body segment and each crutch [50]; and (ii) the diameter of the handle is approximately 2 cm wider than the diameter of the shaft; hence, two markers on the shaft and two markers on the handle would provide a more accurate representation of the vertical and anterior–posterior axis of the crutch. Markers’ trajectories were low-pass filtered with a zero-phase 6th-order Butterworth filter with a cut-off frequency of 5 Hz. A local reference frame was defined on the crutch aligning the axis with the shaft and the handle of the crutch. 

With reference to Figure 9, we derived a rotation matrix R gl and a translation vector T g between the laboratory reference frame, defined by means of the markers placed on the crutch, and the mCrutch reference frame of the embedded IMU. Given the rotation matrices $R_{g}$ of the crutch in the laboratory reference frame and $R_{l}$ the rotation matrix of the mCrutch (embedded IMU) reference frame, we match the origins of the reference systems to their centroids, and we calculate $R_{T}$ as reported in Equation (1).
(1)$$R_{T}{=R}_{g}{\;R}_{l}{}^{T}$$

The singular value decomposition (SVD) method [51] is applied to $R_{T}$ as shown in Equation (2).
(2)$$\left[U,\;D,\;V\right]{=\mathrm{svd}\;(R}_{T})$$

Outcomes of the SVD, namely U, S, and D, which are, respectively, the left singular vectors, the singular values, and the right singular vectors of the homogeneous transformation matrix T, are used to calculate the rotation matrix R gl as shown in Equation (3).
(3)R gl=U [10001000det (U VT)] VT

The latter was also used in Equation (4) for calculating the translation vector T g, where $G_{m}$ and ${\;L}_{m}$ correspond to the centroids of $R_{g}$ and $R_{l}$, respectively.
(4)T g=Gm−R gl Lm

Finally, in Equation (5), R gl and T g are applied to $R_{l}$, to obtain the calibrated rotation matrix, $R_{c}$, which is then converted to Euler angles.
(5)Rc=R gl Rl+T g

The uniaxial load cell in the smart tip requires calibration too. Data obtained from the force platform are used as the gold standard. The load cell calibration is expressed in Equation (6).
(6)$$\hat{F}=k\cdot F+F_{0}$$
where:

$\hat{F}$   is the calibrated value of the force measurement;

F   is the measured value from smart tip;

$F_{0}$ is the offset compensation factor;

k    is the gain factor.

## 3. Results

Preliminary results indicate that the mCrutch system can provide accurate measurements for both pitch and roll angles and applied force. 

Figure 10 illustrates an example of force recorded for both the left and right crutch during the calibration procedure. A random sequence of movements and applied loads, in the range of ±50° and 0–400 N, respectively, were performed for approximately 120 s of recording while the crutches were placed on the force platform (FP). 

In the example, after the calibration, the measured force $\hat{F}$ shows, for the right crutch, an RMSE (root mean square error) < 10 N and a median difference of 2.8 N with an applied force range of approximately 400 N, while for the left crutch, it shows an RMSE < 5 N and a median difference of 1.6 N with an applied force range of approximately 200 N.

Regarding the roll and pitch angles of the crutches, Figure 11 shows an example of rotations around the anterior–posterior (AP) and medio–lateral (ML) axes of the crutch after the calibration with the SVD method. The accuracy in dynamic conditions shows an RMSE < 5° for rotations about the ML axis and an RMSE < 4° for rotations about the AP axis.

## 4. Discussion

This study aimed to describe the mCrutch prototype architecture and report on the accuracy of orientation angles and applied force. From an aesthetic point of view, aside from the status LED and the power button, the instrumented crutches are equivalent to any simple pair of Lofstrand crutches. mCrutch can stream data at a sampling frequency of 100 Hz and the WiFi connectivity with TCP protocol ensures data retransmission. Under normal usage, the mCrutch app requires only 5% of the total CPU usage and approximately 78 MB of memory, with light energy consumption as defined in Android Studio Profiler. This ensures that the app runs smoothly without causing any significant drain on the smartphone’s resources or battery life. Concerning Arduino’s computational time and complexity, sensor reading is not computationally expensive, unlike onboard orientation extraction. This feature requires a high-performance MCU, in particular for a data rate such as 100 Hz. RP2040 has the required computational capacity, thanks to its computational power enabled by a Dual-Core Arm Cortex-M0+ operating at a frequency up to 133 MHz. 

With reference to Section 1.2, different technologies and mounting options have been proposed in the literature. Crutches instrumented with: (i) strain-gauge bridges and a tri-axial accelerometer attached to the external frame and streaming data to a workstation equipped with LabVIEW [21,22]; (ii) hand grip and tilt sensors with data streaming to a workstation equipped with LabVIEW [20] strain gauges mounted on the handle, and an IMU in an external case attached to the frame and streaming data on a laptop [26]; and different types of sensorized tips to measure the applied force for rehabilitation purposes always streaming data to a PC [23,24,25,27,28]. Applications were always limited to an indoor supervised setting. 

mCrutch is an mHealth system prototype, it embeds all electronics inside the frame of the crutch, and it is ready to be scaled up for mass production. It enables both indoor and outdoor scenarios and real-time biofeedback applications implemented as mobile apps. Enabling outdoor applications, especially in unsupervised settings, can open many new lines of research since, as for gait without walking aids, gait patterns observed in a controlled environment may differ from those observed in a real-life setting [52].

The possibility to monitor crutch usage, also in terms of applied force and correct positioning, in real-life scenarios is a first step toward offering new telemonitoring and telerehabilitation services for supporting continuity of care approaches. mCrutch can be connected to personal health systems for personalized interventions and to enhance patient adherence to rehabilitation protocols.

Results show that the calibrated data for both orientation angles and applied force are in line with the results reported in the literature for similar applications [23,25,29]. Regarding the applied force, we decided not to include any information on the crutch orientation in the force estimate to avoid introducing an additional source of error due to the accuracy of the orientation estimate itself. Since partial weight bearing is between 30% to 50% of a patient’s body weight [53], mCrutch shows an error of about 2.5% of the measurement range. As an example, assuming a male person weighing 90 kg, the applied weight on the crutches may range from 27 kg to 45 kg, (approximately 270 N to 450 N). With a 2.5% error, we obtain an error in the force estimate that is in the range of 7 to 11.5 N.

The position of the crutches while standing and walking plays an essential role in rehabilitation; the monitoring and correction of incorrect usage/positioning of the crutches can lead to more personalized and efficient rehabilitation protocols [54,55]. The calibrated pitch and roll angles show an RMSE < 5°. With reference to Figure 11, it is possible to notice relatively large local errors between mocap and mCrutch, e.g., at 20 s and 35 s for both AP and ML planes in the figure. These local errors have a significant impact on the overall RMSE. Possible explanations for these local differences can be the quality of the IMU raw signals that are input to the Madgwick filter and the poor reliability of the Arduino board when it comes to relatively high performance and high accuracy applications. The choice for the Arduino RP2040 board was only due to the very limited volume available inside the crutch handle; to the best of our knowledge, that board was the only one on the market with a width of <2 cm. 

One of our aims was to keep manufacturing costs as low as possible to target mass-market applications. While it would have been possible to enhance the accuracy of the mCrutch sensing components quite easily by employing high-performance load cell [56] and 6-axis IMUs [57,58], this would have increased costs by 5 to 10 factors. The typical cost of a set of crutches is approximately EUR 50, and usage time varies a lot depending on the target group, from a few months for an orthopedic patient to many years for some neurological conditions. Since the cost of the crutches in Italy, and in many other healthcare services, is borne by the patient themselves, the aim is to keep costs as low as possible. To the best of our knowledge, no instrumented crutches are currently available on the market, and the prototypes reported in the literature (Section 1.2) do not report the manufacturing costs and do not address the scaling up of the system in general. We assume that by keeping the final cost for the end-users well below a factor 10 with respect to the typical Lofstrand crutches we can make the system accessible to most patients. 

The absence of a magnetometer in the 6-axis IMU used for the current mCrutch prototype does not allow a robust estimation of the angles about the vertical axis (yaw) [59,60]; hence, the focus was to establish the accuracy of the estimate for the pitch and roll angles. Another possible issue in the estimate of the yaw angle is the crutch’s metallic structure, which can impact the magnetometer’s reliability. The reliability and the added value of the yaw estimate will be addressed in future prototype releases.

As the introduction outlines, crutches are assistive devices used in various neurodegenerative and orthopedic pathologies that may impact walking ability and mobility in general. More specifically, crutches allow patients to rely on a larger support base or better distribute the load between impaired and non-impaired limbs and decrease the load applied to the lower limbs. The accuracy of the current prototype allows bio-feedback and telemonitoring applications with relatively low resolution, but clinically useful thresholds must be established for biofeedback design and for the monitoring of patients’ progress in rehabilitation programs. The suitability or the need to improve the system’s accuracy will be investigated in future studies for specific clinical use cases and patient populations. 

Further studies will also be needed to establish the usability and clinical validity of this mobile health approach. Aside from the need for reliable and accurate measurements, in both supervised and unsupervised settings, battery lifetime is often a critical factor. To maximize system performance and reliability, we opted for TCP WiFi connectivity that requires a much higher battery capacity with respect to Bluetooth low-energy connectivity but provides a much higher performance and reliability. One of the major limitations of the current prototype is the battery life which is now limited to approximately 4 h of continuous data streaming and the real-time processing mode. This battery life would be suitable for rehabilitation sessions but not long-term monitoring applications; hence, the current version does not meet this objective. In order to address this limitation and to improve the accuracy and usability of the system in general, we will design a custom electronic board to replace the Arduino board. This would, in any case, be a necessary step for the scaling up of the system and for ensuring full control over the manufacturing process. We plan to introduce a tilt sensor and relatively large local storage on the new board to enable/disable the sampling of the sensors and to enable/disable the data streaming when real-time feedback is not required. These enhancements would enable both training and long-term monitoring applications with a battery life of a whole day or longer. Another planned improvement of the prototype is adding a third sensing unit placed on the patient’s trunk to monitor his/her posture and the relative orientation of the trunk and the crutches while walking and during other daily activities. Further development of the mCrutch app will support real-time biofeedback modalities and options as well as dedicated user interfaces for both therapists and patients. As stated before, the current version was designed and is only suitable for researchers and developers. The current version of the app allows users to start and stop the data collection, verify the data streams while holding the smartphone and looking at the interface, and test the crutches while keeping the smartphone in their pocket. An app designed for the patient would require automatic pairing with the instrumented crutches, automatic connection of the crutches to the smartphones, and user-friendly configuration panels.

## 5. Conclusions

This study aimed to present mCrutch, a low-cost smart device suitable for mobile health applications in the continuity of care scenarios. The system is composed of a set of instrumented crutches and a smartphone app and allows the collecting and processing in real-time orientation angles and applied loads. The current prototype shows an accuracy of approximately 10 N RMSE for the applied force and 5° for pitch and roll angles which allows bio-feedback and telemonitoring applications that require relatively low resolution, although clinically useful thresholds are yet to be established. Current and expected manufacturing costs for mCrutch are reasonable for a personal device, such as a walking aid. The suitability and clinical validity of mCrutch will be investigated in future studies for specific target populations and settings.

## Figures and Tables

**Figure 1 sensors-23-04151-f001:**
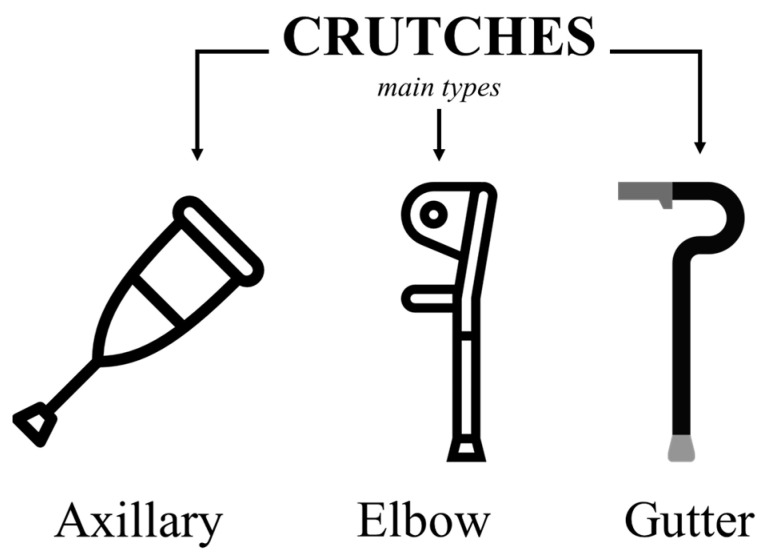
Main types of crutches.

**Figure 2 sensors-23-04151-f002:**
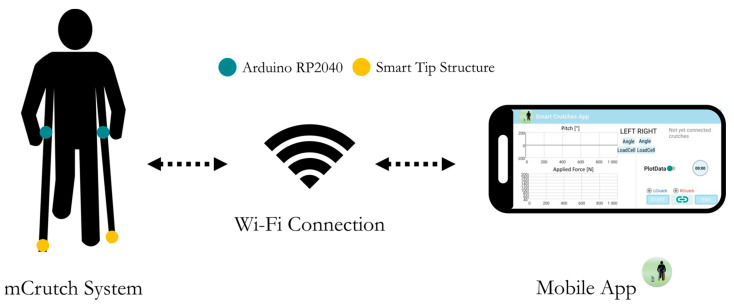
System overview: mCrutches and mobile app.

**Figure 3 sensors-23-04151-f003:**
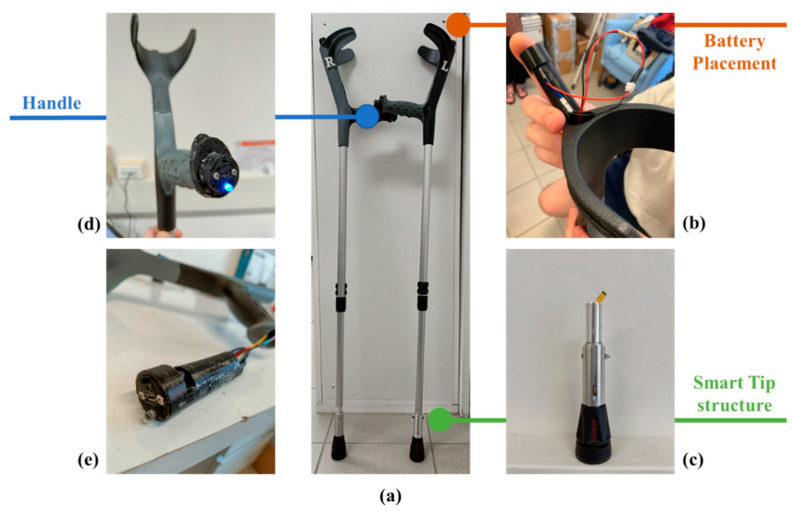
mCrutch assembly: (**a**) mCrutch final prototype; (**b**) battery placement; (**c**) tip modified mechanical structure embedding the miniaturized load cell; (**d**) plastic protective case containing the electronics; and (**e**) handle’s front panel with the status LED, the power button, and the USB port for battery charging.

**Figure 4 sensors-23-04151-f004:**
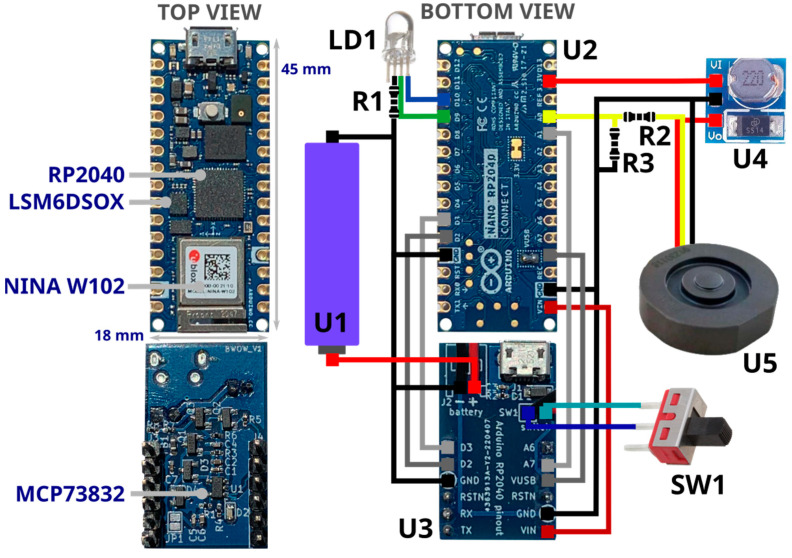
Diagram of electronic connections.

**Figure 5 sensors-23-04151-f005:**
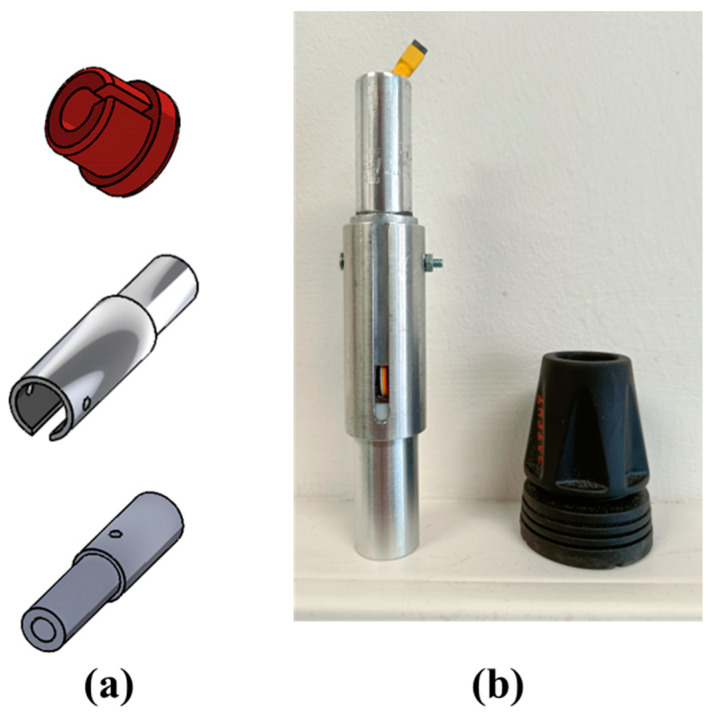
Smart tip structure: (**a**) CAD models and (**b**) finished structure.

**Figure 6 sensors-23-04151-f006:**
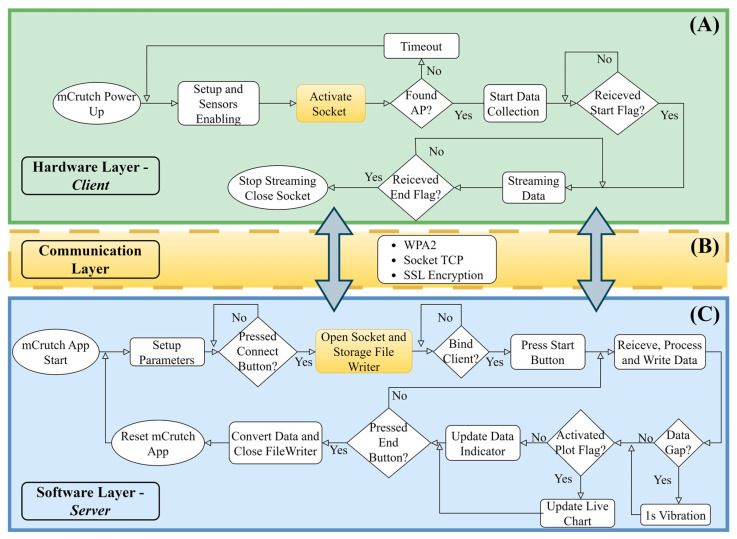
Layers of the mCrutch system architecture. The figure shows the flowcharts of the Arduino/data collection layer (**A**), the communication layer (**B**), and the mCrutch app layer (**C**).

**Figure 7 sensors-23-04151-f007:**
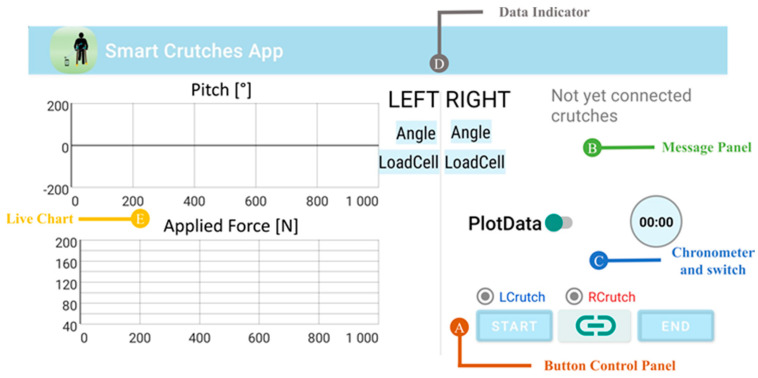
Smart Crutches app layout and descriptors: (**A**) button control panel; (**B**) message panel from system or mCrutch; (**C**) chronometer and switch for a real-time plot; (**D**) data indicator for both pitch [°] and applied force [N]; (**E**) live-chart of pitch [°] in upper graph and applied force [N] in the lower graph.

**Figure 8 sensors-23-04151-f008:**
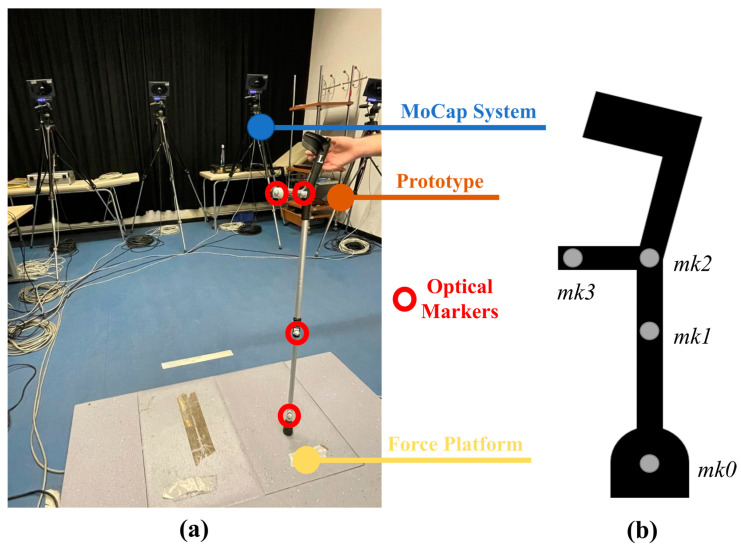
Calibration setup: laboratory environment (**a**) and marker positioning (**b**).

**Figure 9 sensors-23-04151-f009:**
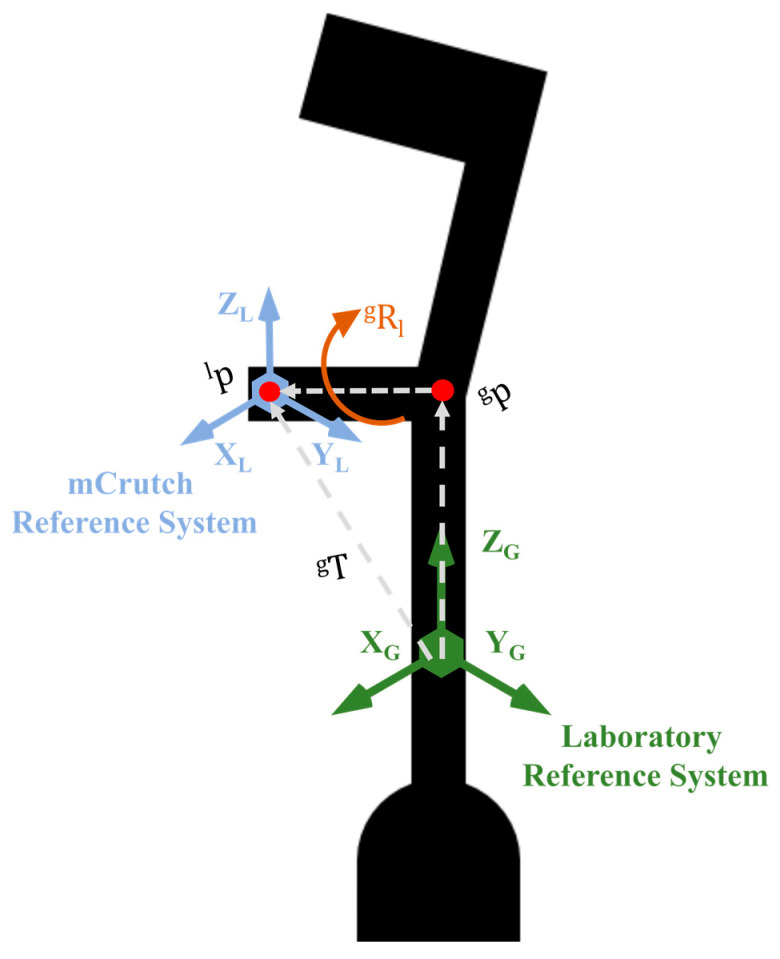
Relationship between the mCrutch reference frame and the laboratory reference frame.

**Figure 10 sensors-23-04151-f010:**
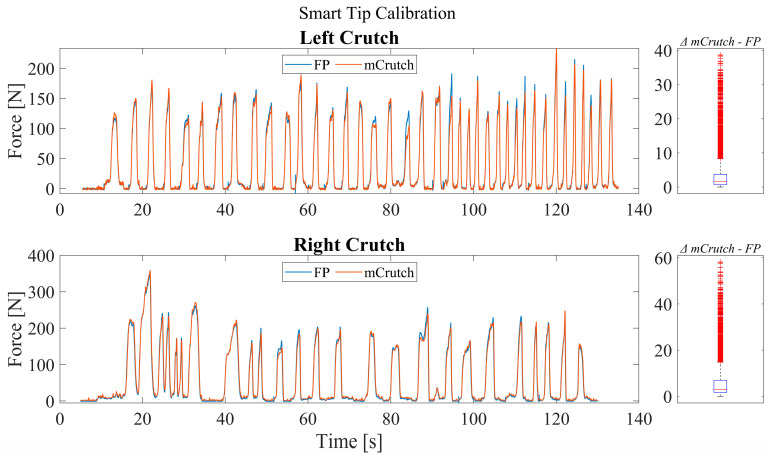
Smart tip calibration performance and distribution of the difference between FP and mCrutch.

**Figure 11 sensors-23-04151-f011:**
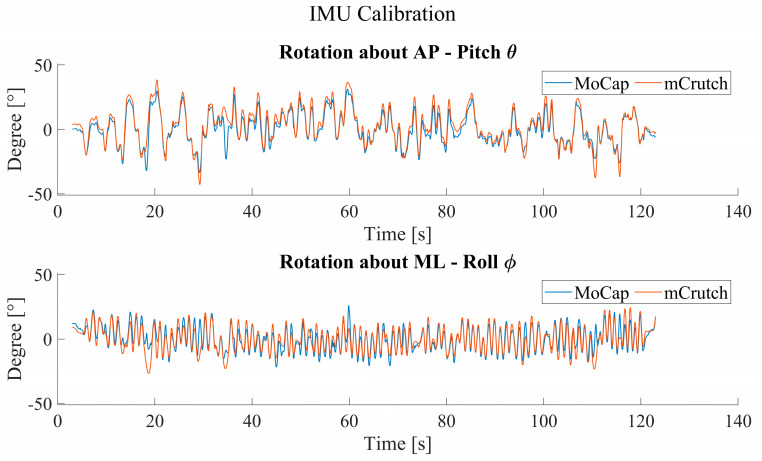
Orientation calibration performance.

**Table 1 sensors-23-04151-t001:** mCrutch electronic characteristics summary.

Features	Description
Computational power	Dual-core 32-bit ARM up to 133 MHz, 264 KB SRAM
Connectivity	Wi-Fi 802.11b/g/n
Orientation estimation	6-axis IMU (accelerometer + gyroscope)
Applied force	Load cell full scale: 500 N
Power supply	Li-Ion battery, 3.7 V at 820 mAh (charger on-board, up to 500 mAh)
LED indicator	Green: power on Blue: connected to smartphone

## Data Availability

The mCrutch system presented in this paper is available for research projects and academic collaborations upon request by contacting the corresponding author.

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
