# Peer review of "mCrutch: A Novel m-Health Approach Supporting Continuity of Care"

_sensors, 2023, doi:10.3390/s23084151_

Round 1

Reviewer 1 Report

This study reports on a smart crutch that can measure the movement of the crutch and the force applied to the crutch by attaching an IMU or force sensor to the crutch in order to quantitatively evaluate the movement of a person using the crutch. The paper makes the case for a cost-effective device configuration that can be used by a large number of people. While we believe these efforts are very important, this paper does not make sufficient claims of originality or quantitative evaluation of cost compared to previous studies.For example, what were the problems of the conventional smart crutches and how did authors solved them. If the system or the performance were the problem, target value and the author's idea for the solution must be shown. Similaly, if the costs were the problem the most costly point and solution must be shown. 

There are also many points in the current paper that are lacking for the discussion.

First, are both mk0 and mk1 attached to the cane used to measure cane posture? I consider that one or the other should be sufficient. Any comments would be appreciated.

Next, regarding the experiment in Figure 9, which compares the magnitude of the force measured by the crutch with the output of the force plate, how was the force measured in the experiment? Were the force measurements made with the cane in a constant posture? Or were the force measurements taken with the cane in various positions as it is normally used? If the posture of the cane is different, then there may be a specific relationship between the force measurements and the posture, but without knowing the posture of the cane, it is difficult to evaluate and or discuss the reason of this difference.
Regarding the experiment in Figure 10, there are places where there is a large gap between the posture measured by the IMU sensor and these results measured by motion capture (for example, at the 35 sec of the pitch angle). Why is this difference in measurement occurring? As same the case of Figure 9, I think it is necessary to compare the posture of the cane and the measurement accuracy in addition to continuous measurement, such as whether there is any angular dependence.

Also, as mentioned above, line 421 discusses the cost of the cane, but it does not provide any logical proof why does the crutch costs only 50 Euros and is this calculation is reasonable or not.

Reviewer 2 Report

sensors-2313528

The authors of the manuscript “mCrutch: a novel m-Health approach supporting continuity of care” developed a mCrutch using smart technologies to collect data to improve the patient's rehabilitation process. Therefore, I found the topic is relevant and important in the field of biomedical engineering. Saying that I have the following points which I would like the authors to consider:

  1. The schematic diagram for the system architecture required more detailed information and a table of the specifications presented on page six lines (203-229).
  2. Provide a flowchart of the data collection by Arduino and a flowchart for the data receiving.
  3. Provide a flow chart for the layered architecture with implementation details.
  4. What about the security of the information? Please provide information about the DataTransmitter security and privacy.
  5. The concept is good; however, this article requires a valid comparison with other mHealth devices.
  6. Lines 242-246 required more explanation regarding any data collection disturbance.

Overall, it is a good article, and I enjoyed reading it. The topic is relevant to the journal and will benefit the readers of the sensor journal. 

Reviewer 3 Report

The paper addresses an interesting topic. It is also nicely written. I have some other comments as follows.

(1) In the abstract, please mention the main quantitative result.

(2) Please report the computational complexity of the system.

(3) Please discuss the real-time aspect of the system.

(4) Please elaborate the 'low cost' of the system. Please compare the proposed system with other systems in terms of low cost. 

Reviewer 4 Report

Submitted article reports the architecture for smart crutches system for mobile health application with the goal of supporting continuity of care for patient using the crutches and long-term monitoring. The paper is well-written and technically sound in what it presents. The prototype consists of three parts:
 - Mechanical: crutches and Smart Tip
 - Electrical: Arduino and accompanying components
 - Software: Android App
which are well documented and designed system is compact and fits in the frame of the crutches.

The authors claim that this is the first system for long-term monitoring, but the authors also said that the presented design is not suitable for long-term monitoring. Presented design battery life is around 4h while in the cited literature similar designs with longer battery life can be found. Thus, the long-term monitoring claim is not justified.

While the proposed Mobile App is useful in some cases for data acquisition and data monitoring, for long-term monitoring the App seems impractical in its current form. Constant wireless data transmission from the sensor to the phone increases power consumption and reduces the battery life. Thus, was it
better to store data in the local memory and then gather it in bursts instead of continuous streaming? On the other hand, if the patient uses both crutches it will be almost impossible to use the phone during that time. From that, it seems that the Mobile App can be used by patient only to gather data and the plotting option is unnecessary. Or the mobile app is intended to be used by the medical personnel and not by the patient. The computing capability of mobile devices is mentioned, but no signal processing and signal analysis is performed. Moreover, there are no technical details about the App itself. Please consider open sourcing your application code on GitHub or similar place and referencing that in the article.

The authors did not provide error comparison of their measurement with the related existing system measurements. I suggest that they use the right axis of Figures 9 and 10 to plot the difference of blue and red lines. For example in Figure 10 it can be seen that for rotations around the AP axis the error around 35s is quite large. The same can be seen for ML axis at around 20s. These discrepancies between MoCap and mCrutch app should be clarified.

My largest concern is that the paper seems more as a patent application or an advertising material such as white paper than the scientific one. The presented material is technically fairly shallow. However, since the paper is for the Special Issue which it fits, I believe that making an elaborate table which will list all similar instrumented crutches (basically the ones mentioned in the subsection 1.2 of the article) and compare them against each other and with your work in detail would suffice.

A minor note is that it should be an mHealth instead of a mHealth, since the pronunciation is "em health".

Round 2

Reviewer 4 Report

The authors are ignoring my requests, but it's fine for me to accept the article anyways.